# Evaluating chatbots in psychiatry: Rasch-based insights into clinical knowledge and reasoning

Yu Chang[1,2], Si-Sheng Huang[3,4], Wen-Yu Hsu[2,3,4], Yi-Chun Liu[3,4,5,6]*

1 Department of Psychiatry, Chung Shan Medical University Hospital, Taichung, Taiwan, 2 School of Medicine, Chung Shan Medical University, Taichung, Taiwan, 3 Post Baccalaureate Medicine, National Chung Hsing University, Taichung, Taiwan, 4 Department of Psychiatry, Changhua Christian Hospital, Changhua, Taiwan, 5 Department of Psychiatry, Changhua Christian Children's Hospital, Changhua, Taiwan, 6 Department of Health Business Administration, Hungkuang University, Taichung City, Taiwan

* 183711@cch.org.tw

## Abstract

Chatbots are increasingly being recognized as valuable tools for clinical support in psychiatry. This study systematically evaluated the clinical knowledge and reasoning of 27 leading chatbots in psychiatry. Using 160 multiple-choice questions from the Taiwan Psychiatry Licensing Examinations and Rasch analysis, we quantified performance and qualitatively assessed reasoning processes. OpenAI's ChatGPT-o1-preview emerged as the top performer, achieving a Rasch ability score of 2.23, significantly surpassing the passing threshold ($p < 0.001$). While it excelled in diagnostic and therapeutic reasoning, it also demonstrated notable limitations in factual recall, niche topics, and occasional reasoning biases. Our findings indicate that while advanced chatbots hold significant potential as clinical decision-support tools, their current limitations underscore that rigorous human oversight is indispensable for patient safety. Continuous evaluation and domain-specific training are crucial for the safe integration of these technologies into clinical practice.

## Introduction

Chatbots, powered by generative artificial intelligence and trained through deep learning algorithms, are designed to engage in natural conversations [1]. These systems, often referred to as large language models (LLMs), exhibit a remarkable ability to process and respond with contextually relevant information [1]. Recent advancements in large-scale data training and sophisticated reasoning mechanisms have expanded chatbots' capabilities from general knowledge dissemination to specialized applications [2]. In the medical field, research has demonstrated that chatbots can achieve acceptable levels of performance in various professional examinations and assessments [3–8].

The growing interest in chatbots for clinical support highlights their potential to enhance healthcare delivery [9]. Previous studies have established a connection

**Data availability statement:** Restrictions apply to the availability of these data. Data(licensing questions) were obtained from the Taiwanese Society of Psychiatry and are available at https://www.sop.org.tw/news/n_list.asp. Otherwise, the data underlying the results presented in the study are available from https://www.kaggle.com/datasets/yuchangsnes5503/evaluating-chatbots-in-psychiatry/data.

**Funding:** The author(s) received no specific funding for this work.

**Competing interests:** The authors have declared that no competing interests exist.

between clinical competence and foundational knowledge [10], suggesting that an evaluation of chatbots' clinical knowledge could provide valuable insights into their practical utility. While previous research has used Rasch analysis to assess chatbots' knowledge in psychiatry, gaps in understanding remain, particularly in certain specialized areas [4]. A comprehensive evaluation of chatbots' strengths and limitations in psychiatric clinical knowledge is still needed.

Measuring psychiatric clinical knowledge poses unique challenges due to the complexity and context-dependent nature of psychiatric assessments. Past research has attempted to evaluate chatbots in this domain [6,11,12], with one study administering a set of ten multiple-choice questions focused on differential diagnosis to both chatbots and psychiatrists [6]. However, the study primarily relied on surface-level comparisons of scores and lacked sufficient depth to quantify or qualitatively understand the underlying knowledge structures of the chatbots. This limited scope has left aspects of chatbot capabilities underexplored, such as their reasoning processes.

To address these limitations, our study employs Rasch analysis [13], a statistical method commonly used in educational and psychological testing to quantify item difficulty and respondent ability on a unified scale. By applying Rasch analysis to a larger and more diverse set of psychiatry-related questions, we aim to provide a detailed assessment of the strengths and weaknesses of the most advanced chatbots. Our study focuses on exploring these foundational aspects to provide insights that can guide their effective integration into clinical practice.

## Materials and methods

### Study design and question selection

To evaluate the clinical knowledge of chatbots, this study utilized multiple-choice questions (MCQs) from the Taiwan Psychiatry Licensing Examination administered by the Taiwanese Society of Psychiatry. To comprehensively evaluate chatbot performance, we included questions related to any clinically relevant topic based on all levels outlined in Bloom's Taxonomy [14]. Specifically, we selected MCQs from the Taiwan Psychiatry Licensing Examination from the past two years (2023 and 2024). These exams represent the first stage of obtaining board certification and consist entirely of MCQs. Questions from these exams are primarily derived from the 12th edition of the Kaplan & Sadock's Synopsis of Psychiatry, ensuring that the content reflects the latest psychiatric knowledge. These questions were crafted by experienced board-certified psychiatrists, and each exam consisted of 100 questions, each worth one point. A score of 60 is set as the passing threshold. To maintain relevance and consistency, questions involving Taiwan-specific laws and policies or those solely testing basic medical knowledge were excluded. This exclusion ensures the broader applicability of our findings, as legal and regulatory frameworks vary across regions. Our study focuses on evaluating core psychiatric knowledge and reasoning rather than jurisdiction-specific legal aspects. The remaining questions were categorized into six domains: pathophysiology and epidemiology, diagnostic assessment and clinical examination, psychopharmacology and other therapeutic modalities, psychosocial and cultural influences, neuroscience and behavioral science, and forensic psychiatry and ethic.

## Chatbot selection

We selected chatbots based on their rankings from the LMarena website [15], which provides comparative evaluations of multiple chatbots. This platform allows users to input queries and compare responses from two chatbots, with the final rankings determined by aggregated user choices. The top 10 ranked chatbots were initially screened in this study, excluding those with restricted availability (e.g., geofenced in mainland China). This process identified leading chatbot companies for inclusion. The final selection comprised chatbots from OpenAI, Google, Anthropic, Meta, xAI, Alibaba, and Mistral. We included all available models from each company to ensure diversity in capabilities and performance. The chatbots evaluated in this study included OpenAI's GPT-4o, GPT-4o-mini, GPT-o1-preview, GPT-o1-mini, and GPT-4-Turbo; Google's Gemini-1.5-Pro, Gemini-1.5-Flash, Gemini-1.5-Flash-8B, Gemini-Exp1121, and LearnLM1.5 Pro; Anthropic's Claude-2, Claude-2.1, Claude-3-Haiku, Claude-3-Sonnet, Claude-3-Opus, Claude-3.5-Haiku, Claude-3.5-Sonnet, and Claude-3.5-Sonnet-June; Meta's Llama-3.1-70B, Llama-3.1-405B, Llama-3.1-Nemotron, and Llama-3.2-90B; xAI's Grok-beta; Alibaba's Qwen-2 and Qwen-2.5; and Mistral's Large-2.

## Evaluation procedure

The evaluation was conducted from November 29 to December 1, 2024. In this study, a zero-shot testing approach was employed, meaning that the chatbots were presented with questions without any prior examples, demonstrations, or contextual information related to the test items. Each chatbot underwent testing with batches of 10 MCQs. The standardized prompt provided to the chatbots was: "Below are 10 multiple-choice questions with their options. Please provide the answers as numbers only." Limiting each batch to 10 MCQs ensured that all questions fit within the context length constraints of the chatbots. Each chatbot's responses were recorded, and the correctness of each answer was documented. For a deeper understanding of chatbot reasoning, we further prompted the chatbots to explain their answers for specific questions using the standardized query: "Please explain your reasoning for Question X in detail." This use of uniform prompts was essential to ensure comparability across models by minimizing performance variations that can arise from different phrasing and batching strategies to which chatbots are sensitive.

The primary outcomes of this study were twofold: (1) the raw score for each chatbot, which was subsequently converted into a logit ability estimate using the Rasch model, and (2) a qualitative assessment of the strengths and weaknesses of the best-performing chatbot. The latter focused on its explanations for individual questions, with particular attention to instances where it correctly answered difficult questions or incorrectly answered simple ones. We assessed the explanations in three aspects: (1) Factual Accuracy: Was the explanation based on correct clinical and pharmacological facts? (2) Logical Coherence: Did the reasoning follow a logical path from premises to conclusion? (3) Identification of Nuance and Bias: Did the model grasp the core clinical principle being tested and avoid common reasoning errors?

## Statistic analysis

First, we performed descriptive analysis, including calculations of the mean, standard deviation, maximum and minimum scores, and Cronbach's alpha to assess the internal consistency of the test. We then conducted further analysis using the Rasch model, a statistical method widely used in psychometrics to evaluate the relationship between item difficulty and the ability of respondents. Rasch analysis is based on modern test theory and provides estimates of item difficulty and respondent ability on the same scale, expressed in logits (log-odds units). Since there is no absolute measure of question difficulty, we employed the Rasch model to assess the relative difficulty levels of the selected questions. This approach also facilitates future comparisons with other medical licensing examinations beyond Taiwan, providing a broader reference for chatbot assessment. The analysis was performed using the WINSTEPS software (version 5.8.2), which is a leading commercial software about Rasch analysis, on a Windows 10 operating system.

In the Rasch model, the probability of a chatbot answering a question correctly is determined by the difference between the chatbot's ability (β) and the item's difficulty (δ) [16]. This is calculated using the formula: $P(X = 1) = \frac{e^{\beta - \delta}}{1 + e^{\beta - \delta}}$, where $P(X = 1)$ represents the probability of a correct response. When the chatbot's ability (β) matches the difficulty of the item (δ), the probability of answering correctly is 50%. The Rasch model iteratively adjusts these estimates to minimize error, producing reliable measures of both ability and difficulty.

Before conducting the Rasch analysis, we first checked whether the dataset met the model's basic assumptions. This involved using principal component analysis (PCA) on the residuals to ensure that the test mainly measured one underlying concept. In Rasch analysis, it's important that secondary dimensions (other factors besides the main concept) explain no more than 20% of the residual variance [4,17]. We also calculated essential unidimensionality, which shows how much of the variance is explained by the main concept—in this case, psychiatric clinical knowledge. A value of 50% or higher was considered acceptable [18]. After confirming unidimensionality, we estimated chatbot ability and item difficulty using joint maximum likelihood estimation (JMLE). The passing threshold was set at 60% accuracy for the selected questions and was also estimated using JMLE. To evaluate whether the best-performing chatbot significantly surpassed this threshold, we performed Wald tests, considering a two-tailed p-value of < 0.05 as statistically significant.

To ensure the model's validity, we analyzed fit statistics to determine if the data aligned with the expectations of the Rasch model. Infit mean square (MNSQ) was utilized to evaluate the consistency of responses to items matching the chatbot's ability level, while outfit MNSQ detected unusual response patterns to extremely easy or difficult items. Acceptable values for both metrics ranged from 0.5 to 1.5 [19]. Additionally, z-standardized (ZSTD) was evaluated, with acceptable values between ±1.96 [19].

Finally, we created a person–item map (PKMAP) for each chatbot, providing a visual representation of its performance across questions of varying difficulty and showing how its ability matches the difficulty levels of the test items. We conducted a detailed analysis of its responses to identify patterns (as mentioned in the evaluation procedure section), with particular focus on the simplest questions it answered incorrectly and the most challenging ones it answered correctly.

## Results

### Chatbot performance overview

Table 1 listed the distribution of questions across two years of licensing exams, comprising 160 questions divided into six categories. The majority of questions were categorized under Pathophysiology and Epidemiology, Diagnostic Assessment and Clinical Examination, and Psychopharmacology and Other Therapeutic Modalities. Table 2 lists the chatbots analyzed in this study, along with their release dates and associated companies. These chatbots were released between July 2023 and November 2024. We evaluated the performance of 27 chatbots (Table 3), yielding an average raw score of 97.7 (61% accuracy), with a standard deviation of 19.5. The highest score was 129, while the lowest was 56. The test reliability Cronbach's alpha was 0.93.

### Dimensionality and Rasch model analysis

The dimensionality analysis confirmed that the dataset aligned with the assumptions of the Rasch model. The raw variance explained by the measures was 38.9%, while the essential unidimensionality calculated as the proportion of

**Table 1. Distribution of question characteristics.**

| Exam Year | Number of Questions | Pathophysiology and Epidemiology | Diagnostic Assessment and Clinical Examination | Psychopharmacology and Other Therapeutic Modalities | Psychosocial and Cultural Influences | Neuroscience and Behavioral Science | Forensic Psychiatry and Ethics |
|---|---|---|---|---|---|---|---|
| 2023 | 80 | 23 | 22 | 22 | 6 | 5 | 2 |
| 2024 | 80 | 22 | 22 | 21 | 4 | 9 | 3 |
| Total | 160 | 45 | 44 | 43 | 10 | 14 | 5 |

**Table 2. Chatbot characteristics.**

| Model Name | Provider | Released Month |
|---|---|---|
| ChatGPT-4o-latest | OpenAI | 2024/11 |
| Gemini-1.5-Pro | Google | 2024/9 |
| Gemini-1.5-Flash | Google | 2024/9 |
| Gemini-1.5-Flash8B | Google | 2024/9 |
| LearnLM1.5 Pro | Google | 2024/11 |
| Claude-3.5-Sonnet | Anthropic | 2024/10 |
| GPT-4o | OpenAI | 2024/05 |
| 4o-mini | OpenAI | 2024/07 |
| o1-preview | OpenAI | 2024/09 |
| o1-mini | OpenAI | 2024/09 |
| Gemini-Exp1121 | Google | 2024/11 |
| Claude-3.5-Sonnet-June | Anthropic | 2024/06 |
| GPT-4-Turbo | OpenAI | 2023/11 |
| Mistral-Large-2 | Mistral | 2024/07 |
| Claude-3-Sonnet | Anthropic | 2024/03 |
| Claude-3-Haiku | Anthropic | 2024/03 |
| Claude-3-Opus | Anthropic | 2024/03 |
| Qwen2-72B | Alibaba | 2024/06 |
| Claude-2 | Anthropic | 2023/07 |
| Claude-2.1 | Anthropic | 2023/11 |
| Qwen-2.5-72B | Alibaba | 2024/09 |
| Claude-3.5-Haiku | Anthropic | 2024/10 |
| Llama-3.2-90B | Meta | 2024/09 |
| Llama-3.1-70B | Meta | 2024/07 |
| Llama-3.1-405B | Meta | 2024/07 |
| Llama-3.1-Nemotron | Meta | 2024/09 |
| Grok-beta | xAI | 2024/11 |

Rasch-common variance, reached 76.5%. These values exceed the commonly accepted thresholds of 20% for unidimensionality and 50% for essential unidimensionality, supporting the interpretation that the observed response patterns primarily reflect a single latent trait as psychiatric clinical knowledge.

The Rasch model parameters for chatbots were presented in Table 3. ChatGPT-o1-preview achieved the highest performance among all models, with a JMLE ability score of 2.23, substantially surpassing the passing threshold (JMLE = 0.44, $p < .001$). Its infit (MNSQ = 1.08, ZSTD = 0.54) and outfit (MNSQ = 0.97, ZSTD = 0.07) statistics were within the acceptable range of 0.5–1.5 and ±1.96, respectively, indicating strong consistency with the Rasch model's expectations. The chatbot's responses demonstrated both reliability and validity in assessing psychiatric clinical knowledge.

## Performance and reasoning of ChatGPT-o1-preview

The person–item map (PKMAP) for ChatGPT-o1-preview (Fig 1) visually represents its performance across questions of varying difficulty levels. In the PKMAP, the vertical axis represents the difficulty of questions in logits, with more difficult items located higher on the map, while the horizontal axis separates correct responses (on the left) from incorrect responses (on the right). The upper-left quadrant (e.g., Items 52, 54, 79, and 140) of the map highlights areas where ChatGPT-o1-preview demonstrated strong capabilities, successfully answering challenging questions. Conversely, the

**Table 3. Raw scores and Rasch model parameters for chatbots.**

| Model Name | Raw score | JMLE measure | JMLE SE | Infit MNSQ | Infit ZSTD | Outfit MNSQ | Outfit ZSTD |
|---|---|---|---|---|---|---|---|
| ChatGPT-4o-latest | 117 | 1.47 | 0.24 | 0.8912 | −0.7891 | 0.9468 | −0.0991 |
| Gemini-1.5-Pro | 117 | 1.47 | 0.24 | 0.7862 | −1.6592 | 0.6234 | −1.4494 |
| Gemini-1.5-Flash | 110 | 1.1 | 0.22 | 1.1754 | 1.4112 | 1.1662 | 0.7412 |
| Gemini-1.5-Flash8B | 77 | −0.33 | 0.2 | 1.1408 | 1.6011 | 1.3981 | 1.6614 |
| LearnLM1.5 Pro | 117 | 1.47 | 0.24 | 1.0241 | 0.221 | 0.8744 | −0.3691 |
| Claude-3.5-Sonnet | 116 | 1.41 | 0.23 | 0.6761 | −2.7093 | 0.4657 | −2.3495 |
| GPT-4o | 120 | 1.64 | 0.24 | 0.7869 | −1.5692 | 0.7419 | −0.8193 |
| 4o-mini | 76 | −0.38 | 0.2 | 1.0415 | 0.511 | 0.9184 | −0.2891 |
| o1-preview | 129 | 2.23 | 0.27 | 1.0829 | 0.5411 | 0.9724 | 0.071 |
| o1-mini | 77 | −0.33 | 0.2 | 1.1953 | 2.1812 | 2.4051 | 4.6324 |
| Gemini-Exp1121 | 116 | 1.41 | 0.23 | 0.7307 | −2.1893 | 0.4838 | −2.2395 |
| Claude-3.5-Sonnet-June | 115 | 1.36 | 0.23 | 0.7599 | −1.9592 | 0.687 | −1.2193 |
| GPT-4-Turbo | 107 | 0.95 | 0.22 | 1.0189 | 0.201 | 1.0926 | 0.4711 |
| Mistral-Large-2 | 56 | −1.2 | 0.21 | 1.3743 | 3.5814 | 3.0949 | 4.2431 |
| Claude-3-Sonnet | 89 | 0.15 | 0.2 | 0.9344 | −0.6891 | 0.9041 | −0.4291 |
| Claude-3-Haiku | 79 | −0.25 | 0.2 | 1.2249 | 2.4712 | 1.7242 | 2.8217 |
| Claude-3-Opus | 106 | 0.9 | 0.22 | 0.8854 | −0.9991 | 1.0068 | 0.101 |
| Qwen2-72B | 111 | 1.15 | 0.23 | 1.1809 | 1.4312 | 1.3014 | 1.2013 |
| Claude-2 | 67 | −0.74 | 0.2 | 0.9869 | −0.119 | 0.991 | 0.051 |
| Claude-2.1 | 71 | −0.58 | 0.2 | 1.0733 | 0.8711 | 1.1703 | 0.7412 |
| Qwen-2.5-72B | 110 | 1.1 | 0.22 | 0.7036 | −2.6893 | 0.5247 | −2.3395 |
| Claude-3.5-Haiku | 96 | 0.45 | 0.21 | 0.8675 | −1.3491 | 0.6945 | −1.6493 |
| Llama-3.2-90B | 77 | −0.33 | 0.2 | 1.1292 | 1.4811 | 1.1476 | 0.7011 |
| Llama-3.1-70B | 78 | −0.29 | 0.2 | 1.2029 | 2.2512 | 1.3116 | 1.3613 |
| Llama-3.1-405B | 102 | 0.72 | 0.21 | 0.9241 | −0.6791 | 0.775 | −1.0892 |
| Llama-3.1-Nemotron | 94 | 0.36 | 0.21 | 0.9505 | −0.479 | 0.8934 | −0.4891 |
| Grok-beta | 108 | 1 | 0.22 | 0.7748 | −2.0392 | 0.5969 | −1.9794 |

JMLE, Joint maximum likelihood estimation; SE, standard error; MNSQ, mean square; ZSTD: z-standardized.

lower-right quadrant (e.g., Items 27, 36, 42, 43, 77, 92, 106, 117, 131, and 146) indicates areas where the chatbot struggled, with incorrect answers to relatively easier questions. This distribution provides a clear visualization of the chatbot's strengths and weaknesses in its performance on the questions.

A detailed analysis of the chatbot's answering process (Table 4, Table 5) revealed key strengths and weaknesses of its reasoning. ChatGPT-o1-preview excelled in areas such as diagnostic reasoning and broader therapeutic concepts (e.g., recognizing paraphilic disorders and treatment paradigms for schizophrenia), and pharmacological principles (e.g., drug mechanisms, indications, and side effects). However, it exhibited notable limitations in recalling specific factual details (e.g., remission timelines for transvestic disorder, concordance rates for generalized anxiety disorder in twin studies, or diagnostic definitions of negative symptoms). Additionally, biases in reasoning were observed, such as overemphasizing lithium's efficacy in depression augmentation therapy while underestimating the role of antipsychotics or dismissing hypnosis as a therapeutic option. The chatbot also demonstrated its capacity for self-correction. In several cases (e.g., Items 27, 42, 77, 131, and 145), it revised its initial incorrect answers upon re-evaluation, ultimately producing accurate solutions.

```
Hard items answered correctly  —Harder— Hard items answered incorrectly
---------------------------------------------------------------------
|                                          6                          |
|                                          |                          |
|                                          | 2.0  18.0  29.0  45.0  148.0 |
|                                          |                          |
|                                          5                          |
|                                          |                          |
|                                          |                          |
| 79.1                                     | 12.0  31.0  32.0  88.0   |
|                                          4                          |
|                                          |                          |
|                                          |                          |
| 52.1  140.1                              | 13.0  33.0  61.0  136.0  |
|                                          |                          |
| 74.1                                     3                          |
|                                          |                          |
|                                          | 83.0  97.0               |
|------------------------------------------|--------------------------|
| 47.1                                 XXX 59.0  86.0                  |
|-119.1--130.1------------------------------2-40.0--------------------|
| 10.1   22.1  122.1  158.1                |                          |
| 123.1                                    | 92.0  146.0              |
| 142.1  144.1                             | 77.0                     |
|                                          | 36.0  43.0               |
| 23.1   82.1  133.1  135.1                |                          |
| 3.1  147.1                               | 42.0  132.0              |
| 26.1   48.1  91.1  111.1  114.1          | 106.0                    |
| 124.1  134.1                             :                          |
| 14.1   17.1  39.1  81.1  159.1           | 107.0                    |
| 101.1  112.1  118.1  120.1  137.1        |                          |
| 141.1                                    :                          |
| 28.1   60.1  90.1  125.1               0 131.0                      |
| 1.1    57.1  89.1  94.1  100.1  138.1  | 117.0                      |
| 150.1                                    :                          |
| 65.1   69.1  70.1  87.1  96.1  98.1    | 27.0                       |
| 99.1  109.1  113.1  116.1  128.1         :                          |
| 154.1                                    |                          |
| 34.1   41.1  63.1  71.1  72.1  115.1   |                          |
| 127.1                                    :                          |
| 19.1   25.1  35.1  46.1  51.1  56.1    |                          |
| 66.1   75.1  102.1  104.1  105.1       :                          |
| 139.1  155.1                             |                          |
|                                         —1                          |
| 4.1    6.1  16.1  85.1  93.1  108.1    |                          |
| 121.1  153.1                             :                          |
| 11.1   20.1  103.1  110.1  126.1       | 145.0                      |
| 151.1  160.1                             :                          |
| 9.1   53.1  73.1  95.1  129.1  143.1   |                          |
| 156.1                                    :                          |
|                                         —2                          |
| 5.1   15.1  21.1  37.1  38.1  44.1     |                          |
| 55.1   62.1  76.1  149.1                 :                          |
| 58.1   68.1  78.1  84.1  152.1         —3                          |
| 157.1                                    :                          |
|                                         —4                          |
| 7.1    8.1  24.1  30.1  49.1  50.1     |                          |
| 54.1   64.1  67.1  80.1                  :                          |
|                                         —5                          |
---------------------------------------------------------------------
 Easy items answered correctly  —Easier— Easy items answered incorrectly

             Each row is 0.2 logits
```

**Fig 1. The person–item map (PKMAP) of ChatGPT-o1-preview.** It illustrated the relationship between the chatbot's ability and the difficulty of the test items. The vertical axis, measured in logits, represents the difficulty level of the questions.

**Table 4. Strengths analysis of ChatGPT-o1-preview's answering process.**

| Question | Answering Process | Analysis |
|---|---|---|
| 52 | Correctly identified the limited evidence for pharmacological treatment in personality disorders, emphasizing appropriate options for different subtypes (e.g., low-dose antipsychotics for paranoid personality disorder). | The chatbot accurately recognized the nuances of pharmacological interventions for personality disorders, a field with limited evidence, showcasing its strong grasp of clinical concepts. |
| 54 | Correctly diagnosed transvestic disorder with fetishism and accurately identified the lack of evidence supporting serotonin-targeting drugs for fetishistic disorders. However, it incorrectly stated that the DSM-5-TR's complete remission criteria for this condition is two years. | Demonstrated a deep understanding of paraphilic disorders, though misinterpreted DSM-5-TR's complete remission criteria, indicating room for improvement in grasping specific textual details. |
| 79 | Appropriately recommended clozapine for treatment-resistant schizophrenia and highlighted the risks of early treatment discontinuation. | Showcased strong knowledge of schizophrenia management, emphasizing evidence-based therapeutic guidelines for early intervention and long-term maintenance therapy. |
| 140 | Correctly identified that ramelteon is metabolized primarily by CYP1A2 and not CYP3A4, and recognized prazosin's clinical utility in treating PTSD-related nightmares. | Exhibited advanced pharmacological knowledge of ramelteon and prazosin mechanisms and applications. |

DSM-5-TR, Diagnostic and Statistical Manual of Mental Disorders, Fifth Edition, Text Revision; PTSD, Post-traumatic stress disorder.

**Table 5. Weaknesses analysis of ChatGPT-o1-preview's answering process.**

| Question | Answering Process | Analysis |
|---|---|---|
| 36 | Incorrectly assumed Petersen's criteria were universally accepted as international standards, leading to a flawed conclusion. | Highlighted a misunderstanding of MCI diagnostic frameworks, reflecting the chatbot's limitation in distinguishing specific recognized guidelines. |
| 43 | Correctly identified quetiapine as a second-line treatment for GAD but incorrectly prioritized buspirone as a first-line option. | Demonstrated a nuanced understanding of GAD treatment options but relied on an inaccurate assumption about buspirone's role, reflecting a need for improved precision in specific pharmacological hierarchy. |
| 92 | Failed to accurately recognize negative symptoms as specifically outlined in the DSM-5-TR's core definitions. | Struggled with aligning its interpretation to DSM-5-TR's precise wording, indicating a challenge in processing specific textual details and narrowing down to exact definitions. |
| 106 | Incorrectly dismissed hypnosis as a therapeutic tool, while overlooking more evident inaccuracies in alternative approaches—particularly considering that effective treatment does not always require deeply exploring traumatic experiences. | Displayed bias in evaluating hypnosis, possibly due to limitations in its training data, leading to an overly critical stance that ignored significant errors in competing choices. |
| 107 | Failed to recognize that PET scans show decreased basal ganglia and white matter metabolism in GAD patients. Additionally, it inaccurately stated the twin concordance rate for GAD as 15%, which is underestimated. | Revealed knowledge gaps in neuroimaging findings for GAD and difficulties with rare, less commonly reinforced statistical data, such as twin concordance rates for GAD. |
| 117 | Overemphasized lithium's benefits while underestimating antipsychotics' augmentation role and misinterpreted esketamine's sigma receptor action. | Highlighted limitations in less frequently encountered and highly specific pharmacological knowledge, as well as biases in treatment prioritization. |
| 132 | Correctly noted that depressive patients emphasize cognitive issues, but conflated this with caregivers' ability to notice functional impairments. | Demonstrated difficulty in interpreting subjective clinical observations but showed competence in differentiating psychopathology during diagnosis. |
| 146 | Correctly identified that deep sleep decreases with age but struggled to choose the most appropriate answer for the question. | Struggled to prioritize between two valid options but demonstrated a solid understanding of geriatric sleep changes. |

MCI, mild cognitive impairment; GAD, generalized anxiety disorder; DSM-5-TR, Diagnostic and Statistical Manual of Mental Disorders, Fifth Edition, Text Revision; PET, positron emission tomography.

The symbol "XXX" indicates the chatbot's estimated ability, and each item on the map represents a specific question number from the exam, accompanied by either a "1" for correct responses or a "0" for incorrect ones. A "1" indicates a correct response and is displayed on the left side of the map, while a "0" signifies an incorrect response and is placed on the right side. The vertical positioning of each item reflects its difficulty.

## Discussion

To our knowledge, this study is the first to apply Rasch analysis to evaluate chatbots' clinical knowledge and reasoning in psychiatry using expert-designed multiple-choice questions. Rasch model may simplify certain aspects of clinical reasoning complexity, it provides a robust framework for quantifying chatbot performance and identifying specific strengths and weaknesses. A fundamental understanding of chatbots' internal knowledge and reasoning is essential for advancing their clinical applications and scalability. Among the 27 chatbots assessed, ChatGPT-o1-preview emerged as the top-performing model, achieving a correct response rate of 80.6% and a JMLE ability estimate of 2.23. According to the Rasch model, this indicates that ChatGPT-o1-preview would achieve a correct response rate of 85.6%. This performance not only surpassed the passing threshold by a significant margin but also placed its score well within the range of successful human candidates seeking board certification. The strengths of ChatGPT-o1-preview were particularly evident in its understanding of high-level diagnostic, therapeutic, and pharmacological concepts. For instance, the model showcased advanced reasoning in areas such as schizophrenia management across various stages, diagnostic clarity in paraphilic disorders, and a thorough understanding of psychopharmacology, including drug mechanisms, indications, and side effects. Moreover, ChatGPT-o1-preview exhibited a strong ability to self-correct during re-evaluation.

Despite these strengths, the chatbot demonstrated notable limitations. It struggled with questions requiring precise factual recall, such as the exact Diagnostic and Statistical Manual of Mental Disorders, Fifth Edition, Text Revision (DSM-5-TR) criteria [20], remission timelines for transvestic disorder, and rare statistical data (e.g., concordance rates for generalized anxiety disorder). This highlights the need for caution regarding the chatbot's susceptibility to hallucinations or confabulations (a term used when a model produces factually incorrect statements with great confidence, analogous to neuro-psychiatric confabulation) [21]. When probed for highly specific details, ChatGPT-o1-preview occasionally generated inaccurate or fabricated information. This limitation is consistent with prior studies on chatbot performance in factual recall tasks. For example, the SimpleQA benchmark [22], which focuses on short, fact-seeking queries from diverse aspects, requires each question to meet strict criteria: it must have a single, indisputable answer that is easy to grade, and the answer should remain constant over time. Chatbots tend to perform poorly on SimpleQA tasks. Similarly, the Chinese SimpleQA [23], a localized version of the SimpleQA benchmark, has demonstrated similar challenges in chatbot performance. Additionally, the chatbot occasionally displayed reasoning biases, such as overemphasizing lithium's efficacy in depression augmentation therapy while underestimating the role of antipsychotics or dismissing hypnosis as a viable treatment for dissociative identity disorder.

Our findings highlight how biases in medical reasoning, such as the overemphasis on lithium for major depressive disorder augmentation therapy, could pose clinical risks. Identifying these domain-specific biases is crucial, as they may directly impact patient safety. Such biases may stem from the training data's inherent limitations or uneven exposure to certain clinical concepts [24–26]. Such findings align with previous research, including biasmedQA, which highlights how biased information in prompts can lead to biased clinical judgments by chatbots [27]. As for ethical considerations, adhering to the principle of 'do no harm' remains a fundamental criterion in psychiatric care. This principle is especially critical given the unique vulnerabilities of psychiatric patients. For patient populations with compromised reality testing, such as individuals with psychotic disorders or severe cognitive impairment, a single piece of misinformation from an AI could reinforce a delusion, undermine therapeutic trust, or precipitate a clinical crisis. Ensuring accountability for AI-related harm requires robust regulatory frameworks. To mitigate potential errors and their impact on patient care, it is essential to implement strategies such as human oversight, model interpretability, and real-time validation in clinical environments.

Additionally, clinicians must be equipped with guidelines to effectively assess and intervene when chatbot-generated recommendations deviate from best practices.

Our study underscores the importance of both training data quality and ongoing performance optimization in enhancing chatbot reliability for psychiatric applications. While ChatGPT-o1-preview demonstrated superior knowledge in psychiatry, its performance in niche areas was sometimes biased and less reliable. This discrepancy reflects the accessibility of training resources; open-access journals, books, and widely available materials likely dominate the training corpus, while proprietary textbooks and leading psychiatric journals remain less accessible. Recent partnerships between publishers and chatbot developers signal potential solutions to this issue. For instance, Axel Springer has partnered with OpenAI to integrate journalism with AI technologies [28], and Elsevier Health has collaborated with OpenEvidence to develop ClinicalKey AI [29]. Collaborative efforts such as these could bridge the gap between accessible and proprietary knowledge, benefiting chatbot training. Techniques like retrieval-augmented generation (RAG) [30,31], which uses vectorized data to enhance responses by retrieving relevant information, or fine-tuning [32,33], which optimizes models by training them on domain-specific datasets, can further improve chatbot recall ability in specialized fields.

From a clinical perspective, the findings suggest that chatbots like ChatGPT-o1-preview hold significant potential as tools for delivering timely and relatively accurate feedback to support clinical decision-making. While chatbots, even multimodal or more advanced models, may hold potential for clinical use, they cannot yet be directly integrated into psychiatric practice without rigorous experimental validation and regulatory considerations. Human oversight remains indispensable to ensure safe and effective implementation. First, given the chatbot's performance limitations, it is advisable to present it with pre-defined clinical options and guide it to provide explanations, rather than relying on it to generate responses independently. Employing adequate prompt engineering [34,35], which involves designing specific and structured queries, can substantially improve the quality of chatbot responses. Second, encouraging chatbots to perform iterative reasoning or re-evaluate their outputs has been shown to enhance accuracy, as demonstrated in this study. Third, leveraging multiple models simultaneously can provide diverse perspectives and help mitigate biases inherent in individual systems. Finally, rather than depending on chatbots for factual recall, they are better suited for reasoning and decision support, where their outputs can effectively complement clinical expertise. By adopting a thoughtful and cautious approach, chatbots can serve as valuable adjuncts to enhance patient care while minimizing risks.

The rapid advancements in chatbot capabilities cannot be ignored. Historical research suggests that as of early 2024, chatbots had achieved performance sufficient to pass licensing exams, with some models, such as ChatGPT-o1-preview, significantly surpassing that benchmark [4]. This accelerated improvement suggests that conclusions from earlier studies, particularly those conducted in 2023 or earlier [36,37], may no longer accurately reflect the current state of chatbot capabilities. Therefore, ongoing evaluation of new chatbot is essential to keep pace with their evolving capabilities and to better understand their applicability in clinical practice.

## Limitations

This study has several limitations. First, the sample size of 27 chatbots, although sufficient for a pilot study using Rasch analysis [38], may limit the generalizability of our findings. However, the large number of questions used (n = 160) ensured a low standard error for ability estimates, enhancing the reliability of the results. Additionally, our focus on state-of-the-art (SOTA) models may limit the generalizability of our findings to smaller or less advanced language models. Second, while the multiple-choice questions used in this study provided a standardized method of assessing psychiatric knowledge, they may not fully capture the breadth of clinical expertise required in practice. Nevertheless, establishing a clear understanding of chatbots' knowledge base and reasoning processes remains a fundamental step before integrating them into AI-assisted clinical decision-making, especially given the potential risks associated with their deployment. Third, all questions were presented in traditional Mandarin, and the results may vary across different languages. However, core psychiatric knowledge and reasoning are largely language-independent, as fundamental clinical principles remain consistent across

different linguistic contexts. Fourth, while possessing clinical knowledge is important, it does not necessarily translate to the effective application of that knowledge in real-world scenarios. Although we have highlighted key aspects of potential clinical application, future research should prioritize systematically test for biases to ensure reliability and evaluating the efficacy of chatbots in supporting clinical decision-making through controlled trials in real-world psychiatric settings [39–41].

## Conclusion

This study demonstrates that ChatGPT-o1-preview, released by OpenAI in September 2024, outperformed other chatbots in a standardized evaluation of psychiatric clinical knowledge. The model's strengths lie in its understanding of diagnostic frameworks, treatment paradigms, and pharmacology concepts. However, its limitations in recalling specific details, addressing niche knowledge, and overcoming biases highlight the need for cautious interpretation of its outputs. Building on these findings, we have proposed key aspects of practical considerations to support their integration into psychiatric practice. As chatbot technologies continue to evolve, ongoing assessments of their capabilities and controlled clinical trials will be crucial for understanding their broader applicability and ensuring safe and effective implementation in clinical settings.

## Author contributions

**Conceptualization:** Yu Chang, Si-Sheng Huang, Yi-Chun Liu.

**Formal analysis:** Yu Chang, Yi-Chun Liu.

**Methodology:** Yu Chang, Wen-Yu Hsu.

**Software:** Yu Chang.

**Validation:** Yi-Chun Liu.

**Writing – original draft:** Yu Chang, Si-Sheng Huang, Wen-Yu Hsu, Yi-Chun Liu.

**Writing – review & editing:** Yu Chang, Yi-Chun Liu.

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
