## [Decision Letter · Decision Letter 0]

26 Feb 2025

Thank you for submitting your manuscript to PLOS ONE. After careful consideration, we feel that it has merit but does not fully meet PLOS ONE’s publication criteria as it currently stands. Therefore, we invite you to submit a revised version of the manuscript that addresses the points raised during the review process.

We look forward to receiving your revised manuscript.

Kind regards,

Lucija Gosak

Academic Editor

PLOS ONE

Journal Requirements:

3. Please include captions for your Supporting Information files at the end of your manuscript, and update any in-text citations to match accordingly. Please see our Supporting Information guidelines for more information: http://journals.plos.org/plosone/s/supporting-information .

Reviewers' comments:

Reviewer's Responses to Questions

**Comments to the Author**

1. Is the manuscript technically sound, and do the data support the conclusions?

Reviewer #1: No

Reviewer #2: Yes

2. Has the statistical analysis been performed appropriately and rigorously?

Reviewer #1: No

Reviewer #2: I Don't Know

3. Have the authors made all data underlying the findings in their manuscript fully available?

Reviewer #1: Yes

Reviewer #2: Yes

4. Is the manuscript presented in an intelligible fashion and written in standard English?

Reviewer #1: No

Reviewer #2: Yes

Reviewer #1: The study provides an assessment of the utility of chatbots in psychiatric clinical practice. The research evaluates 27 chatbots using 160 multiple-choice questions derived from the Taiwan Psychiatry Licensing Examination. Employing the Rasch model for statistical analysis, the study offers quantitative insights into the ability of chatbots to handle psychiatric clinical knowledge and reasoning. The standout model, ChatGPT-o1-preview, displayed exceptional capabilities in diagnostic and treatment reasoning and psychopharmacology, with a Joint Maximum Likelihood Estimation (JMLE) ability score of 2.23.

Overall, the paper needs substantial improvements and changes, as explained below

1. The study's findings are narrowly applicable, focusing solely on chatbots tested in traditional Mandarin. This language limitation reduces the relevance and applicability of the results for global audiences, particularly in non-Mandarin-speaking regions.

2. The study evaluates chatbot performance using standardized multiple-choice questions (MCQs), which do not adequately capture the complexities and nuances of real-world psychiatric clinical practice. There is no evidence of chatbot evaluation in live clinical settings or through practical scenarios.

3. While the study evaluates 27 chatbots, this is a relatively small sample size considering the rapidly evolving landscape of AI-driven language models. The lack of diversity among evaluated chatbots raises questions about the robustness and generalizability of the findings.

4. The study highlights significant limitations in factual recall and reasoning biases among chatbots but does not propose robust methods for mitigating these issues. This undermines the reliability of the conclusions and fails to address critical concerns about safety in clinical settings.

5. While the Rasch model is a well-established psychometric tool, its application here simplifies the complexity of clinical knowledge and reasoning. It may not be the most suitable approach to fully assess chatbots' ability to handle nuanced psychiatric scenarios.

6. The exclusion of questions related to Taiwan-specific laws and policies limits the comprehensive evaluation of the chatbots' knowledge. This omission raises concerns about the completeness and depth of the study.

7. The study primarily reaffirms well-known strengths and limitations of chatbots (e.g., strong reasoning, poor factual recall, and biases). It does not provide any groundbreaking insights or propose innovative solutions to address the identified challenges.

8. The study lacks a detailed discussion of ethical implications and safety concerns regarding the deployment of chatbots in psychiatric settings. Furthermore, the manuscript does not explore the impact of potential errors on patient care or how clinicians can effectively mitigate these risks.

9. While the study identifies the potential utility of chatbots in psychiatry, it does not validate these findings through experimental or observational studies in clinical environments. This limits the practical significance of the conclusions.

10. The article's findings largely replicate conclusions from prior studies on chatbot performance in medical fields, offering little novelty or value to the existing body of knowledge.

Reviewer #2: The article is well written.

How was the level of difficulty of the chosen questions from the Taiwan Psychiatry Licensing Examinations assessed to test the range analytic ability of the Chatbots?

What are the limitations of this study?

**Do you want your identity to be public for this peer review?** For information about this choice, including consent withdrawal, please see our Privacy Policy

Reviewer #1: No

Reviewer #2: No

---

## [Author Response · Author response to Decision Letter 1]

2 Mar 2025

Please find the attached document containing our formatted responses.

Reviewer #1: The study provides an assessment of the utility of chatbots in psychiatric clinical practice. The research evaluates 27 chatbots using 160 multiple-choice questions derived from the Taiwan Psychiatry Licensing Examination. Employing the Rasch model for statistical analysis, the study offers quantitative insights into the ability of chatbots to handle psychiatric clinical knowledge and reasoning. The standout model, ChatGPT-o1-preview, displayed exceptional capabilities in diagnostic and treatment reasoning and psychopharmacology, with a Joint Maximum Likelihood Estimation (JMLE) ability score of 2.23.

Overall, the paper needs substantial improvements and changes, as explained below

We sincerely appreciate your time and effort in reviewing our manuscript. Your comments have prompted us to further refine our explanations and clarify certain aspects of our study. We would like to respectfully address your comments. Below, we provide our detailed point-to-point responses.

1. The study's findings are narrowly applicable, focusing solely on chatbots tested in traditional Mandarin. This language limitation reduces the relevance and applicability of the results for global audiences, particularly in non-Mandarin-speaking regions.

Thank you for your insightful comment. We acknowledge that our study is conducted in Traditional Mandarin, which may affect the generalizability of the findings. Meanwhile, core psychiatric knowledge and reasoning are largely language-independent, as fundamental clinical principles remain consistent across different linguistic contexts. While language differences may impact chatbot performance to some extent, the fundamental clinical principles assessed in this study remain broadly relevant across different linguistic contexts. We have clarified this point in the revised manuscript as following: “However, core psychiatric knowledge and reasoning are largely language-independent, as fundamental clinical principles remain consistent across different linguistic contexts.” (Section Limitations, Page 21, Line 345-347)

2. The study evaluates chatbot performance using standardized multiple-choice questions (MCQs), which do not adequately capture the complexities and nuances of real-world psychiatric clinical practice. There is no evidence of chatbot evaluation in live clinical settings or through practical scenarios.

Thank you for your valuable comment. We acknowledge that MCQs differ from real-world clinical settings and do not fully capture the complexities of psychiatric practice. Our study aims to assess chatbots' knowledge and reasoning rather than their direct application in clinical environments. Understanding their knowledge base and reasoning processes is a fundamental step before integrating them into AI-assisted clinical decision-making. Given the potential risks associated with chatbot deployment in psychiatry, it is crucial to first establish a clear understanding of their capabilities and limitations before considering their use in real-world settings. We have clarified this point in the revised manuscript as following: “Nevertheless, establishing a clear understanding of chatbots’ knowledge base and reasoning processes remains a fundamental step before integrating them into AI-assisted clinical decision-making, especially given the potential risks associated with their deployment” (Section Limitations, Page 21, Line 341-344)

3. While the study evaluates 27 chatbots, this is a relatively small sample size considering the rapidly evolving landscape of AI-driven language models. The lack of diversity among evaluated chatbots raises questions about the robustness and generalizability of the findings.

Thank you for your insightful comment. We acknowledge that AI models evolve rapidly, and it is essential to consider this factor when interpreting our findings. To ensure a comprehensive evaluation, we carefully selected the most state-of-the-art (SOTA) models available at the time of our study. Compared to prior research (for example, in citation 5 and 6), which often assesses a smaller number of chatbots, our study includes a relatively broad sample. We recognize that our focus on SOTA models may limit the generalizability of our findings to smaller or less advanced language models. We have clarified this point in the revised manuscript as following, “Additionally, our focus on state-of-the-art (SOTA) models may limit the generalizability of our findings to smaller or less advanced language models.”. (Section Limitations, Page 21, Line 337-339)

4. The study highlights significant limitations in factual recall and reasoning biases among chatbots but does not propose robust methods for mitigating these issues. This undermines the reliability of the conclusions and fails to address critical concerns about safety in clinical settings.

Thank you for your valuable comment. We have revised the discussion section to further elaborate on potential methods for mitigating factual recall limitations and reasoning biases in chatbots, as following: “Our study underscores the importance of both training data quality and ongoing performance optimization in enhancing chatbot reliability for psychiatric applications. While ChatGPT-o1-preview demonstrated superior knowledge in psychiatry, its performance in niche areas was sometimes biased and less reliable.” and “Techniques like retrieval-augmented generation (RAG) [30, 31], which uses vectorized data to enhance responses by retrieving relevant information, or fine-tuning [32, 33], which optimizes models by training them on domain-specific datasets, can further improve chatbot recall ability in specialized fields.” (Section Discussion, Page 19, Line 292-295; Section Discussion, Page 19, Line 303-306).

5. While the Rasch model is a well-established psychometric tool, its application here simplifies the complexity of clinical knowledge and reasoning. It may not be the most suitable approach to fully assess chatbots' ability to handle nuanced psychiatric scenarios.

Thank you for your comment. We acknowledge that the Rasch model is a statistical approach for assessing chatbot performance in a structured and standardized manner. Our study does not claim that the results directly translate to real-world psychiatric scenarios; rather, it focuses on evaluating chatbots' knowledge and reasoning within a controlled setting. While we recognize that the Rasch model may simplify certain aspects of clinical reasoning complexity, it provides a robust framework for quantifying chatbot performance and identifying specific strengths and weaknesses. We have clarified this point in the revised discussion as following, “Rasch model may simplify certain aspects of clinical reasoning complexity, it provides a robust framework for quantifying chatbot performance and identifying specific strengths and weaknesses.” (Section Discussion, Page 17, Line 248-250)

6. The exclusion of questions related to Taiwan-specific laws and policies limits the comprehensive evaluation of the chatbots' knowledge. This omission raises concerns about the completeness and depth of the study.

Thank you for your comment. While clinical knowledge and reasoning are generally universal, legal and regulatory frameworks vary significantly across regions. To ensure the broader applicability of our findings, we intentionally excluded Taiwan-specific laws and policies, as they may not be relevant to chatbot performance in other healthcare systems. This decision aligns with our study’s focus on evaluating core psychiatric knowledge and reasoning rather than jurisdiction-specific legal aspects. We have clarified this rationale in the revised manuscript as following, “This exclusion ensures the broader applicability of our findings, as legal and regulatory frameworks vary across regions. Our study focuses on evaluating core psychiatric knowledge and reasoning rather than jurisdiction-specific legal aspects.” (Section Study Design and Question Selection, Page 4, Line 79-82)

7. The study primarily reaffirms well-known strengths and limitations of chatbots (e.g., strong reasoning, poor factual recall, and biases). It does not provide any groundbreaking insights or propose innovative solutions to address the identified challenges.

Thank you for your comment. While previous studies have explored factual recall limitations and biases in general knowledge, there has been limited investigation into these issues within medical specialties. Our study provides new insights by demonstrating that biases in medical reasoning, such as an overemphasis on lithium for major depressive disorder (MDD) augmentation therapy, could pose potential clinical risks. Identifying these domain-specific biases is crucial, as they may have direct implications for patient safety. We have clarified this contribution in the revised discussion as following, “Our findings highlight how biases in medical reasoning, such as the overemphasis on lithium for major depressive disorder augmentation therapy, could pose clinical risks. Identifying these domain-specific biases is crucial, as they may directly impact patient safety.” (Section Discussion, Page 18, Line 279-281)

8. The study lacks a detailed discussion of ethical implications and safety concerns regarding the deployment of chatbots in psychiatric settings. Furthermore, the manuscript does not explore the impact of potential errors on patient care or how clinicians can effectively mitigate these risks.

Thank you for your comment. We acknowledge that ethical implications and safety concerns are critical aspects of AI deployment in psychiatric settings. While our study does not advocate for the direct clinical use of chatbots at this stage, we recognize the importance of addressing potential risks. We have modified as following, “As for ethical considerations, adhering to the principle of 'do no harm' remains a fundamental criterion in psychiatric care. Ensuring accountability for AI-related harm requires robust regulatory frameworks. To mitigate potential errors and their impact on patient care, it is essential to implement strategies such as human oversight, model interpretability, and real-time validation in clinical environments. Additionally, clinicians must be equipped with guidelines to effectively assess and intervene when chatbot-generated recommendations deviate from best practices.” (Section Discussion, Page 19, Line 285-295)

9. While the study identifies the potential utility of chatbots in psychiatry, it does not validate these findings through experimental or observational studies in clinical environments. This limits the practical significance of the conclusions.

Thank you for your comment. We acknowledge that our findings are not directly validated in clinical environments. Our study aims to provide an initial evaluation of chatbots' psychiatric knowledge and reasoning, which is a necessary step before considering real-world implementation. We have clarified this limitation in the revised discussion as following, “While chatbots, even multimodal or more advanced models, may hold potential for clinical use, they cannot yet be directly integrated into psychiatric practice without rigorous experimental validation and regulatory considerations.”. (Section Discussion, Page 20, Line 309-311)

10. The article's findings largely replicate conclusions from prior studies on chatbot performance in medical fields, offering little novelty or value to the existing body of knowledge.

Thank you for your comment. While prior studies have examined chatbot performance in general medical applications, our study specifically focuses on psychiatric knowledge and reasoning, which has been less explored in existing literature. Unlike previous research, we employ a systematic evaluation using the Rasch model to quantify chatbot performance of psychiatry, providing a more structured and interpretable analysis. Additionally, we identify unique domain-specific biases, such as the overemphasis on lithium for major depressive disorder augmentation, which have not been extensively reported in prior work. We have clarified these contributions in the revised discussion as following, “Our findings highlight how biases in medical reasoning, such as the overemphasis on lithium for major depressive disorder augmentation therapy, could pose clinical risks. Identifying these domain-specific biases is crucial, as they may directly impact patient safety.” (Section Discussion, Page 18, Line 279-281)

Reviewer #2: The article is well written.

Thank you for your positive feedback. We appreciate your affirmation.

How was the level of difficulty of the chosen questions from the Taiwan Psychiatry Licensing Examinations assessed to test the range analytic ability of the Chatbots?

Thank you for your question. The Taiwan Psychiatry Licensing Examination has a passing score of 60, with an estimated passing rate of approximately 85% (though this data is not publicly available). While there is no absolute measure of question difficulty, we employed the Rasch model to assess the relative difficulty levels of the selected questions. This method quantifies item difficulty on a logit scale, allowing for a structured evaluation of chatbot performance across varying levels of complexity. Additionally, our approach enables future comparisons with other medical licensing examinations, including those outside of Taiwan, to provide a broader context for chatbot assessment. We have clarified this methodology in the revised manuscript as following: “Since there is no absolute measure of question difficulty, we employed the Rasch model to assess the relative difficulty levels of the selected questions. This approach also facilitates future comparisons with other medical licensing examinations beyond Taiwan, providing a broader reference for chatbot assessment.” (Section Statistical Analysis, Page 6, Line 125-128)

What are the limitations of this study?

Thank you for your comment. We have added a dedicated limitations section to explicitly discuss the constraints of this study. We have revised manuscript to provide a more comprehensive discussion, as following: “This study has several limitations. First, the sample size of 27 chatbots, although sufficient for a pilot study using Rasch analysis [38], may limit the generalizability of our findings. However, the large number of questions used (n = 160) ensured a low standard error for ability estimates, enhancing the reliability of the results. Additionally, our focus on state-of-the-art (SOTA) models may limit the generalizability of our findings to smaller or less advanced language models. Second, while the multiple-choice questions used in this study provided a standardized method of assessing psychiatric knowledge, they may not fully capture the breadth of clinical expertise required in practice. Nevertheless, establishing a clear understanding of chatbots’ knowledge base and reasoning processes remains a fundamental step before integrating them into AI-assisted clinical decision-making, especially given the potential risks associated with their deployment. Third, all questions were presented in traditional Mandarin, and the results may vary across different languages. However, core psychiatric knowledge and reasoning are largely language-independent, as fundamental clinical principles remain consistent across different linguistic contexts. Fourth, while possessing clinical knowledge is important, it does not necessarily translate to the effective application of that knowledge in real-world scenarios. Although we have highlighted key aspects of potential clinical application, future research should prioritize evaluating the efficacy of chatbots in supporting clinical decision-making through controlled trials in real-world psychiatric settings [7, 39].” (Section Limitations, Page 21, Line 333-352)

---

## [Decision Letter · Decision Letter 1]

25 Jul 2025

Dear Dr. Liu,

Thank you for submitting your manuscript to PLOS ONE. After careful consideration, we feel that it has merit but does not fully meet PLOS ONE’s publication criteria as it currently stands. Therefore, we invite you to submit a revised version of the manuscript that addresses the points raised during the review process.

The manuscript has been evaluated by two reviewers, and their comment is available below. Reviewer 3 has requested minor modifications in various sections of the manuscript.

Could you please carefully revise the manuscript to address the comment raised?

We look forward to receiving your revised manuscript.

Kind regards,

Zahra Al-Khateeb, Ph.D

Staff Editor

PLOS ONE

Journal Requirements:

Reviewers' comments:

Reviewer's Responses to Questions

**Comments to the Author**

Reviewer #3: All comments have been addressed

Reviewer #4: All comments have been addressed

2. Is the manuscript technically sound, and do the data support the conclusions?

Reviewer #3: Yes

Reviewer #4: Yes

3. Has the statistical analysis been performed appropriately and rigorously?

Reviewer #3: Yes

Reviewer #4: Yes

4. Have the authors made all data underlying the findings in their manuscript fully available?

Reviewer #3: Yes

Reviewer #4: Yes

5. Is the manuscript presented in an intelligible fashion and written in standard English?

Reviewer #3: Yes

Reviewer #4: Yes

Reviewer #3: This is a timely and methodologically solid study that evaluates the performance of 27 state-of-the-art chatbots using a large, standardized set of psychiatry board exam questions.

The use of Rasch analysis to assess chatbot performance represents a methodological advancement over prior surface-level comparisons, and the dual quantitative-qualitative approach offers valuable insights into chatbot reasoning.

The manuscript is clearly written, well-organized, and contributes meaningfully to the literature on AI in clinical education and decision support.

Minor Comments:

- Zero-shot prompting is mentioned, but the paper would benefit from a more detailed discussion of prompt design and its implications (e.g., potential effects of different phrasings or batching).

- The qualitative evaluation (Tables 4 and 5) is excellent but could benefit from a more structured rubric or scoring system to increase reproducibility of the “reasoning strength” claims.

- While the authors briefly mention psychiatrist performance in prior studies, adding a comparator (even if drawn from past literature) would help contextualize chatbot scores.

- The term "hallucination" vs. "confabulation" is correctly problematized, but the terminology could be clarified further for a general audience unfamiliar with LLM behavior nomenclature.

- The manuscript acknowledges that no human data were used. However, a deeper reflection on the clinical risks of AI hallucinations in psychiatry—especially in vulnerable populations—would enhance the impact.

- Consider tightening the abstract to make the results and implications more immediately clear.

- Ensure consistent formatting of acronyms (e.g., DSM-5-TR is occasionally formatted inconsistently).

Reviewer #4: Well-written article and discussed the main strengths and limitations of current chatbox use in psychiatry. The future directions are well identified and articulated.

Thank you for outlining these thoughtful mitigation strategies. I particularly agree with the importance of prompt engineering—designing structured, context-rich queries significantly improves chatbot performance. This approach helps guide the model’s reasoning process, minimizes ambiguity, and leads to more clinically relevant responses. It's a practical and effective way to align chatbot outputs with expert expectations, especially in sensitive domains like healthcare.

Thank you for this important work.

**Do you want your identity to be public for this peer review?** For information about this choice, including consent withdrawal, please see our Privacy Policy

Reviewer #3: **Yes: ** Luca Cima

Reviewer #4: **Yes: ** Bernice G. Gulek, PhD, ACNP

---

## [Author Response · Author response to Decision Letter 2]

25 Jul 2025

Reviewer #3: This is a timely and methodologically solid study that evaluates the performance of 27 state-of-the-art chatbots using a large, standardized set of psychiatry board exam questions.

The use of Rasch analysis to assess chatbot performance represents a methodological advancement over prior surface-level comparisons, and the dual quantitative-qualitative approach offers valuable insights into chatbot reasoning.

The manuscript is clearly written, well-organized, and contributes meaningfully to the literature on AI in clinical education and decision support.

We sincerely thank reviewer #3 for the positive assessment and the detailed, thoughtful suggestions for improvement. We have detailed our revisions below.

Minor Comments:

- Zero-shot prompting is mentioned, but the paper would benefit from a more detailed discussion of prompt design and its implications (e.g., potential effects of different phrasings or batching).

Thank you for pointing this out. We have modified the paragraph as following: “The standardized prompt provided to the chatbots was: "Below are 10 multiple-choice questions with their options. Please provide the answers as numbers only." Limiting each batch to 10 MCQs ensured that all questions fit within the context length constraints of the chatbots. Each chatbot’s responses were recorded, and the correctness of each answer was documented. For a deeper understanding of chatbot reasoning, we further prompted the chatbots to explain their answers for specific questions using the standardized query: "Please explain your reasoning for Question X in detail." This use of uniform prompts was essential to ensure comparability across models by minimizing performance variations that can arise from different phrasing and batching strategies to which chatbots are sensitive.” (Section Evaluation Procedure, Page 5-6, Line 104-113)

- The qualitative evaluation (Tables 4 and 5) is excellent but could benefit from a more structured rubric or scoring system to increase reproducibility of the “reasoning strength” claims.

Thank you for pointing this out. We agree that a structured rubric would be the ideal method for ensuring the reproducibility of our qualitative findings. While developing and validating a full quantitative scoring system for chatbot reasoning was beyond the scope of this particular study, we recognize that we can make our analytical process more transparent to address the reviewer's valid point. We concur with the reviewer that developing a formal, expert-validated scoring system for AI reasoning is a critical and promising direction for future research." We have added as following: “We assessed the explanations in three aspects: (1) Factual Accuracy: Was the explanation based on correct clinical and pharmacological facts? (2) Logical Coherence: Did the reasoning follow a logical path from premises to conclusion? (3) Identification of Nuance and Bias: Did the model grasp the core clinical principle being tested and avoid common reasoning errors?” (Section Evaluation Procedure, Page 6, Line 119-123)

- While the authors briefly mention psychiatrist performance in prior studies, adding a comparator (even if drawn from past literature) would help contextualize chatbot scores.

Thank you for pointing this out. We have directly added as following: “According to the Rasch model, this indicates that ChatGPT-o1-preview would achieve a correct response rate of 85.6%. This performance not only surpassed the passing threshold by a significant margin but also placed its score well within the range of successful human candidates seeking board certification.” (Section Discussion, Page 17, Line 261-263)

- The term "hallucination" vs. "confabulation" is correctly problematized, but the terminology could be clarified further for a general audience unfamiliar with LLM behavior nomenclature.

Thank you for your suggestion. We have added clarification as following: “This highlights the need for caution regarding the chatbot's susceptibility to hallucinations or confabulations (a term used when a model produces factually incorrect statements with great confidence, analogous to neuro-psychiatric confabulation).” (Section Discussion, Page 18, Line 275-277)

- The manuscript acknowledges that no human data were used. However, a deeper reflection on the clinical risks of AI hallucinations in psychiatry—especially in vulnerable populations—would enhance the impact.

Thank you for your suggestion. We have supplemented our manuscript as following: “This principle is especially critical given the unique vulnerabilities of psychiatric patients. For patient populations with compromised reality testing, such as individuals with psychotic disorders or severe cognitive impairment, a single piece of misinformation from an AI could reinforce a delusion, undermine therapeutic trust, or precipitate a clinical crisis.” (Section Discussion, Page 19, Line 295-299)

- Consider tightening the abstract to make the results and implications more immediately clear.

Thank you for your suggestion. We have modified and condensed our abstract as following: “Chatbots are increasingly being recognized as valuable tools for clinical support in psychiatry. This study systematically evaluated the clinical knowledge and reasoning of 27 leading chatbots in psychiatry. Using 160 multiple-choice questions from the Taiwan Psychiatry Licensing Examinations and Rasch analysis, we quantified performance and qualitatively assessed reasoning processes. OpenAI's ChatGPT-o1-preview emerged as the top performer, achieving a Rasch ability score of 2.23, significantly surpassing the passing threshold (p < 0.001). While it excelled in diagnostic and therapeutic reasoning, it also demonstrated notable limitations in factual recall, niche topics, and occasional reasoning biases. Our findings indicate that while advanced chatbots hold significant potential as clinical decision-support tools, their current limitations underscore that rigorous human oversight is indispensable for patient safety. Continuous evaluation and domain-specific training are crucial for the safe integration of these technologies into clinical practice.” (Section Abstract, Page 2, Line 18-29)

- Ensure consistent formatting of acronyms (e.g., DSM-5-TR is occasionally formatted inconsistently).

Thank you for catching this inconsistency. We have performed a thorough search of the manuscript and corrected all instances of inconsistent acronym formatting. We will ensure that "DSM-5-TR" and all other acronyms are formatted uniformly throughout the text, tables, and reference list.

Reviewer #4: Well-written article and discussed the main strengths and limitations of current chatbox use in psychiatry. The future directions are well identified and articulated.

Thank you for outlining these thoughtful mitigation strategies. I particularly agree with the importance of prompt engineering—designing structured, context-rich queries significantly improves chatbot performance. This approach helps guide the model’s reasoning process, minimizes ambiguity, and leads to more clinically relevant responses. It's a practical and effective way to align chatbot outputs with expert expectations, especially in sensitive domains like healthcare.

Thank you for this important work.

We are very grateful to reviewer #4 for the positive assessment and encouraging feedback. We are pleased that the reviewer found the article to be well-written and that our discussion of the strengths, limitations, and future directions was well-articulated.

We particularly appreciate the reviewer's validation of our emphasis on mitigation strategies like prompt engineering. This agreement reinforces our conclusion that structured, context-aware interaction is a key method for improving the reliability and clinical relevance of chatbot outputs. It is a critical practical takeaway that we hope our work conveys effectively.

---

## [Editor Report · Decision Letter 2]

30 Jul 2025

Evaluating Chatbots in Psychiatry: Rasch-Based Insights into Clinical Knowledge and Reasoning

PONE-D-25-00276R2

Dear Dr. Liu,

We’re pleased to inform you that your manuscript has been judged scientifically suitable for publication and will be formally accepted for publication once it meets all outstanding technical requirements.

Kind regards,

George Vousden

Staff Editor

PLOS ONE
---

## [Editor Report · Acceptance letter]

PONE-D-25-00276R2

PLOS ONE

Dear Dr. Liu,

I'm pleased to inform you that your manuscript has been deemed suitable for publication in PLOS ONE. Congratulations! Your manuscript is now being handed over to our production team.

Kind regards,

on behalf of

Dr. George Vousden

Staff Editor

PLOS ONE